# Psychometric Characteristics of Smartphone-Based Gait Analyses in Chronic Health Conditions: A Systematic Review

**DOI:** 10.3390/jfmk10020133

**Published:** 2025-04-16

**Authors:** Tobias Bea, Helmi Chaabene, Constantin Wilhelm Freitag, Lutz Schega

**Affiliations:** 1Department of Health and Physical Activity, Institute III, Otto-von-Guericke University Magdeburg, 39104 Magdeburg, Germany; tobias.bea@ovgu.de (T.B.); constantin.freitag@ovgu.de (C.W.F.); lutz.schega@ovgu.de (L.S.); 2Institut Supérieur de Sport et de l’Éducation Physique du Kef, Université de Jendouba, Le Kef 7100, Tunisia

**Keywords:** validity, reliability, feasibility, sensitivity, smartphone, inertial measurement units, gait, patients, chronic diseases, pathological conditions, multiple sclerosis, Parkinson’s disease

## Abstract

**Background:** Chronic health conditions frequently result in gait disturbances, impacting quality of life and mobility. Smartphone-based gait analysis has emerged as a promising alternative to traditional methods, offering accessibility, cost effectiveness, and portability. This systematic review evaluates smartphone-based inertial measurement units’ validity, reliability, and sensitivity for assessing gait parameters in individuals with chronic conditions. **Methods:** A comprehensive literature search in Web of Science, PubMed, Google Scholar, and SportDiscus identified 54 eligible studies. **Results:** Validity was evaluated in 70% of the included studies, with results showing moderate-to-strong associations between smartphone apps and gold-standard systems (e.g., Vicon), particularly for parameters such as gait speed and stride length (e.g., r = 0.42–0.97). However, variability was evident across studies depending on the health condition, measurement protocols, and device placement. Reliability, examined in only 27% of the included studies, displayed a similar trend, with intraclass correlation coefficients (ICCs) ranging from moderate (ICC = 0.53) to excellent (ICC = 0.95) for spatiotemporal parameters. Sensitivity and specificity metrics were explored in 41% and 35% of the included studies, respectively, with several applications achieving over 90% accuracy in detecting gait abnormalities. Feasibility was rated positively in 94% of the included studies, emphasising the practical advantages of smartphones in diverse settings. **Conclusions:** The findings of this systematic review endorse the clinical potential of smartphones for remote and real-world gait analysis, while highlighting the need for standardised methodologies. Future research should adopt a more comprehensive approach to psychometric evaluation, ensuring that reliability aspects are adequately explored. Additionally, long-term studies are needed to assess the effectiveness of smartphone-based technologies in supporting the personalised treatment and proactive management of chronic conditions.

## 1. Introduction

Chronic health conditions (e.g., multiple sclerosis, Parkinson’s disease) often lead to gait disturbances, which can significantly impact mobility, independence, and overall quality of life [1,2]. These gait impairments serve as important clinical markers for assessing disease progression and overall health status [3]. For example, it is well-established that neurodegenerative diseases like Parkinson’s disease and multiple sclerosis can cause motor impairments, leading to slowed walking speed, reduced step length, and increased gait variability [4,5]. Likewise, musculoskeletal disorders such as osteoarthritis and chronic low back pain can result in altered gait mechanics due to pain and joint stiffness, further contributing to mobility restrictions [6,7]. Furthermore, there is evidence that cardiovascular diseases, including peripheral artery disease, have also been associated with gait abnormalities, as reduced circulation can impair muscle function and endurance [8]. These gait impairments result in serious health consequences, as they are associated with adverse outcomes such as increased risk of falls and fractures, loss of independence, and mortality [4,5,6,7]. In this vein, compelling evidence indicates that slower gait speed is a key predictor of frailty and increased mortality risk, while changes in cadence and step length are considered indicators of early neuromuscular or cardiovascular dysfunction [6,7]. For instance, a systematic review and meta-analysis revealed that individuals with persistent low back pain exhibit a significantly slower walking speed and shorter stride length compared to healthy controls, highlighting the functional impact of chronic pain on gait mechanics [8].

Traditionally, gait assessment has relied on sophisticated and expensive technologies, including electromyography [9], motion capture systems (e.g., Vicon) [10], force plates, and instrumented walkways [11], which are considered the gold standard in gait analysis, typically available only in specialised clinical or research settings [12]. While these methods provide accurate measurements of spatiotemporal gait parameters, they require trained personnel and extensive infrastructure, restricting their accessibility and feasibility for widespread clinical use. Moreover, these methods are time-consuming and often impractical for monitoring patients in real-life or home-based environments [13,14]. As such, the need for alternative practical tools has become increasingly evident, particularly during global health crises such as the COVID-19 pandemic, which underscored the importance of remote monitoring technologies.

In recent years, wearable technologies such as inertial measurement units (IMUs), pressure-sensitive insoles, and smartphones have emerged as promising alternative tools [15,16,17]. With embedded sensors, such as accelerometers and gyroscopes, smartphones offer a cost-effective, portable, and accessible solution for capturing motion data [18,19]. These devices enable both real-time and long-term monitoring of gait parameters in natural environments, providing insights that are often more representative of a patient’s functional abilities [20]. Studies in populations such as individuals with Parkinson’s disease, multiple sclerosis, and/or healthy older adults have demonstrated the utility of smartphone-based gait analysis [21,22,23,24,25,26,27]. To ensure that gait analysis methods provide meaningful and reliable data, it is essential to assess their psychometric properties, which include validity and its derivatives, reliability and sensitivity [28,29]. These properties are crucial in determining whether a testing method can be trusted for clinical or research applications. From a broader perspective, validity refers to the extent to which a measurement tool accurately assesses what it is intended to measure [30]. Concurrent validity, a key validity aspect, refers to the degree to which the measurement tool correlates with a gold-standard measure [31]. In gait analysis, this means comparing smartphone-based measurements with established gold-standard systems, such as motion capture or instrumented walkways. A method with poor validity may produce misleading gait abnormality results. Reliability relates to the consistency and repeatability of a measurement method. A tool is considered reliable if it produces stable and consistent results under similar conditions [26]. Sensitivity describes a method’s ability to correctly detect actual gait impairments, while specificity indicates its capability to distinguish between healthy individuals and those with gait abnormalities. High sensitivity ensures that most individuals with gait impairments are correctly identified, reducing the risk of false negatives, whereas high specificity minimises false positives by correctly classifying healthy individuals as non-impaired [32]. In this context, feasibility is a critical factor in determining whether smartphone-based gait analysis can be realistically integrated into routine clinical practice or long-term home monitoring, as technologies with low user compliance or high technical demands may not be sustainable for widespread use [33,34,35].

Earlier systematic reviews have focused on specific populations, such as individuals with Parkinson’s disease or multiple sclerosis [15,36]. The results showed that smartphone applications have great potential to validly assess gait and balance impairments in individuals with these chronic conditions, but that further studies are needed to comprehensively evaluate their reliability and sensitivity [15,36]. Furthermore, comparing young and older adults is essential, as age-related changes in gait patterns influence the interpretation of smartphone-derived gait metrics. Younger individuals typically exhibit more stable gait patterns, whereas older adults, particularly those with chronic conditions, often present with gait variability, increased fall risk, and mobility limitations. Understanding these differences is fundamental for ensuring that smartphone-based gait assessments are valid across diverse populations.

A major gap in the existing literature is the lack of a systematic evaluation of the psychometric properties of smartphone-based gait analysis across different chronic diseases (e.g., Parkinson’s disease, multiple sclerosis, and stroke) and age groups. Without robust psychometric evaluation, the clinical applicability of smartphone-based gait analysis remains uncertain, potentially leading to misleading or non-reproducible results. Therefore, this review aims to fill this gap by critically evaluating the validity, reliability, sensitivity, and feasibility of smartphone-based gait analysis in young and older adults with different chronic conditions. By synthesising the existing evidence, this review seeks to assess the clinical potential of smartphone-based IMUs for gait monitoring, identify limitations in current research, and propose directions for future studies.

## 2. Materials and Methods

The present systematic review was conducted according to the Preferred Reporting Items for Systematic Reviews and Meta-Analyses (PRISMA) guidelines [37]. Additionally, this study has been registered on the Open Science Framework (OSF) at https://doi.org/10.17605/OSF.IO/QFU8D (accessed on 20 December 2024).

### 2.1. Search Strategy

A comprehensive literature search was conducted for studies published between 2014 and 2024, utilising the following electronic databases: Web of Science, PubMed, Google Scholar, and SportDiscus. Only peer-reviewed studies written in English were included. The timeframe chosen reflects an important consideration: prior to 2014, the integration of smartphones and related technologies into daily life was less widespread, limiting their use in general, and specifically in gait analysis. Keywords were collected through expert opinion, literature review, and controlled vocabulary (e.g., Medical Subject Headings [MeSH]). The search was carried out using the Boolean operators “AND” and “OR”. The following is our search strategy applied with PubMed: (“Gait” [MeSH] OR “Gait Analysis” OR “Walking” OR “Locomotion”) AND (“Mobile Applications” OR “Smartphones” OR “Wearable Devices” OR “Mobile Health”) AND (“Pathological Conditions, Anatomical” [MeSH] OR “Neurological Diseases” OR “Musculoskeletal Diseases” OR “Chronic Diseases” OR “Parkinson Disease” OR “Stroke” OR “Multiple Sclerosis” OR “Arthritis” OR “COPD” OR “low back pain”). Additional manual searches of reference lists were performed. Two review authors (TB and CF) conducted the study selection process independently, screening the titles and abstracts of all identified records. Each author compiled a list of potentially eligible studies based on the inclusion criteria. These lists were then compared and discussed to reach a consensus. In cases of discrepancy, a third author (HC) was consulted to ensure an unbiased and rigorous selection process.

### 2.2. Inclusion and Exclusion Criteria

The inclusion criteria for eligible studies were selected based on the PICOS (Population, Intervention, Comparison, Outcome, Study Design) approach. The following criteria were defined: (I), Population: young and old adult participants (age ≥ 18 years) with chronic health conditions (e.g., Parkinson’s disease, multiple sclerosis, musculoskeletal, cardiovascular and metabolic diseases); (II), Intervention: smartphone-based gait analysis using embedded IMUs; (III), Comparison: gold-standard methods for gait analysis (e.g., motion capture systems, force plates) or other validated assessment tools (e.g., trundle wheel or XSens); (IV), Outcome: validity, reliability, sensitivity, and feasibility data; (V), Study design: observational studies, cross-sectional studies, cohort studies or validation studies.

Studies were excluded if they focused solely on healthy participants, involved individuals under 18 years of age, were not in English (except French), and lacked assessments of reliability, validity, sensitivity, and feasibility. The focus on adults (age ≥ 18 years) was chosen to avoid variability arising from maturation changes in younger populations, which are better suited to separate investigations.

### 2.3. Data Extraction and Analysis

From the included studies, detailed data were extracted, including participant demographics such as sample size, age, sex, and underlying chronic conditions. Information about the study design and context, including whether the study was cross-sectional, observational, or a prospective cohort study, was also recorded. Data were systematically gathered on the devices and applications used, such as the smartphone models, application names, intended users, test location, and sensor placement (e.g., sternum or pocket). Key gait parameters, including velocity, step length, and cadence, were noted alongside the gold-standard systems used for validation, such as motion analysis platforms or pressure-sensitive mats. The primary outcomes focused on test–retest reliability, typically reported as intraclass correlation coefficients (ICCs) or typical error of measurement (TEM), and validity metrics, often expressed through correlation or agreement with gold standards. Additionally, feasibility was assessed based on multiple criteria, including user compliance, ease of use, required technical expertise, and the practicality of smartphone-based gait analysis in different environments (e.g., clinical settings vs. home or remote monitoring). Smartphone-based gait analysis tools were considered feasible if they demonstrated high participant adherence, minimal setup complexity, and usability without extensive training. Of note, feasibility evaluation included qualitative assessments from the original studies, such as reported participant feedback and investigator observations on technical difficulties and measurement consistency. Furthermore, sensitivity, specificity, main findings, intended users, and limitations of the studies were extracted. The results of the included studies were qualitatively synthesised. Descriptive analyses were performed to compare reliability and validity outcomes across studies, while variations in device placement and methodologies were highlighted as potential sources of heterogeneity.

### 2.4. Methodological Quality Appraisal

The methodological quality of each included study was assessed using a modified version of the “Quality Assessment Tool for Observational Cohort and Cross-Sectional Studies” originally developed by the National Institute of Health [38]. This tool initially comprises 14 criteria, each rated as “Yes” or “No”. The quality of the studies was classified using clearly defined thresholds: studies with a “Yes” response for at least 8 criteria were categorised as “good quality”, those with 5 to 7 positive assessments were considered “fair quality”, and studies with fewer than 5 positive ratings were classified as “low quality” [39].

## 3. Results

### 3.1. Study Selection

The study selection process and its outcomes are displayed in Figure 1. The search initially identified a total of 946 studies. After removing duplicates, 867 unique studies remained for further evaluation. Screening the titles and abstracts led to the exclusion of 746 studies. Consequently, the full texts of the remaining 121 studies were reviewed. Of these, 45 studies were excluded for not assessing the reliability or validity of gait measurements, 15 studies were excluded due to their focus on healthy participants, and 7 studies were excluded because they involved participants younger than 18 years. Finally, our systematic literature search resulted in 54 studies eligible for inclusion.

### 3.2. Study Analysis

A clear summary of the results from a total of 54 studies investigating smartphone-based applications for gait analysis are displayed in Table 1. A detailed overview of the characteristics of all 54 studies is included in (Appendix A). The number of participants varied significantly across studies, ranging from 10 to 1414 participants. Overall, 70% (38 studies) of the included studies focused on participants with specific health conditions, such as Parkinson’s disease or multiple sclerosis. The remaining 30% (16 studies) analysed either neurological conditions or patients with other conditions, like rheumatoid arthritis. The average age of participants ranged from 21 years to 73 years. Around 29 studies (54%) focused on middle-aged adults (40 to 65 years). Of the 54 studies, 15 studies (28%) were cross-sectional, 11 studies (20%) were observational, 5 (9%) were prospective, 4 (7%) were validation studies, and 3 studies (6%) used longitudinal designs.

Additionally, 52% (28 studies) [35,40,41,42,43,44,45,46,47,48,49,50,51,52,53,54,55,56,57,58,59,60,61,62,63,64,65,66] of the included studies conducted measurements in clinical settings, 37% (20 studies) [22,23,25,40,41,43,57,67,68,69,70,71,72,73,74,75,76,77,78,79] collected data in home environments, and 7% (4 studies) [41,42,43,57] combined both clinical and home settings. Parkinson’s disease was the most commonly studied health condition, with 25 studies (46%) [22,23,25,35,40,44,47,49,52,56,57,60,65,66,67,68,69,71,74,75,76,77,80,81,82], multiple sclerosis was investigated in 10 studies (19%) [41,42,59,70,72,73,78,79,83,84], stroke was the focus of 3 studies (6%) [48,50,58], rheumatoid arthritis was addressed in 2 studies (4%) [64,85], as well as pulmonary diseases (4%) [51,86] and only 1 study focused on chronic low back pain patients (2%) [87]. Overall, the studies primarily investigated smartphone-based gait analysis in patients with neurological (e.g., Parkinson’s disease, multiple sclerosis, stroke) and musculoskeletal (e.g., rheumatoid arthritis, chronic low back pain) conditions. Most studies were conducted in clinical settings (52%), though a significant portion (37%) explored home-based assessments, underscoring the feasibility of smartphone gait monitoring outside of traditional healthcare facilities.

The range of smartphone models used in the studies was diverse, as 14 studies (26%) employed iPhone models, including older devices like the iPhone 4s [43] and newer models like the iPhone 13 [63]; 8 studies (15%) used specialised apps like the mPower app [40,75]; and 16 studies (30%) relied on standard apps such as Google Fit or proprietary applications, as seen in the work from Polese et al. [58]. Sensor placement varied considerably across studies, reflecting ongoing methodological debates regarding optimal positioning. Most of the studies (62%) placed smartphones at the hip or in the pocket, aligning with traditional IMU-based gait assessments. Other studies explored alternative placements, such as the chest (11%), or even less conventional positions like the thigh or sacrum (4%). These variations highlight the lack of standardisation in smartphone-based gait assessments.

Gait speed was measured in 47 studies (87%), including works by Tang et al. [35] and Adams et al. [40]. Step length and stride length were focuses in 32 studies (59%), such as Capecci et al. [44] and Mehrang et al. [75]. Gait variability and fluctuation width were analysed in 16 studies (30%), including Kim et al. [23] and Pepa et al. [57]. Five studies (9%), such as Kim et al. [23], investigated specific parameters like the freezing index. Overall, most studies focused on fundamental spatiotemporal gait parameters, such as gait speed, step length, and cadence, as these are widely used to assess mobility impairments in clinical practice. A smaller subset of studies analysed more complex gait characteristics, such as gait variability, which has been proposed as an early indicator of neurodegenerative disease progression. Some studies specifically targeted disease-specific gait markers, such as the freezing index in Parkinson’s disease.

### 3.3. Validation, Reliability, and Feasibility Outcomes

Table 2 summarises the psychometric properties of smartphone-based gait analysis across various pathological conditions. A detailed summary is attached to (Appendix A). Among the 54 studies included in this review, 70% (38 studies) investigated the validity of smartphone-based gait analysis. Concurrent validity was the most frequently examined aspect, with 35 studies (65%) comparing smartphone-derived gait parameters to gold-standard systems such as motion capture, force plates, instrumented walkways, or wearable IMUs. Correlation coefficients (r) for concurrent validity ranged from 0.42 to 0.97, with higher agreement levels observed for gait speed (r = 0.78–0.97) and cadence (r = 0.74–0.95) compared to step length (r = 0.42–0.90) and gait variability (r = 0.56–0.87). ICC values for agreement varied from 0.53 to 0.96, depending on the gait metric and study design. Concurrent validity was generally higher in clinical settings (r = 0.68–0.92) compared to home-based assessments (r = 0.50–0.81). This trend was also reflected in the broader comparison of measurement conditions, where differences between clinical and home environments were observed in 37% (20 studies). Specifically, clinical studies showed stronger correlations (r = 0.75–0.97) due to standardised test conditions, whereas home-based assessments exhibited greater variability (r = 0.42–0.91), likely influenced by environmental factors such as surface conditions and participant adherence.

Sensitivity and specificity values followed a similar trend, with clinical assessments demonstrating higher diagnostic accuracy (sensitivity 85–98%, specificity 79–97%) compared to home settings (sensitivity 62–91%, specificity 59–94%). Construct validity, more specifically, discriminative validity, was examined in six studies (11%), primarily through subgroup comparisons of healthy and pathological gait patterns. Sensitivity and specificity were reported in 41% and 35% of the studies, respectively. Sensitivity values ranged from 43% to 98%, and specificity from 59% to 97%, with the highest classification accuracy reported for Parkinson’s disease (sensitivity 88–98%, specificity 83–97%). Studies on multiple sclerosis and stroke showed moderate classification performance, with sensitivity values ranging from 62% to 89%. The lowest diagnostic accuracy was observed in musculoskeletal disorders and chronic pain conditions, where gait alterations were less distinct. Among studies focusing on different smartphone models, validation outcomes varied depending on sensor specifications, with higher correlations observed in studies using devices with tri-axial accelerometers and gyroscopes. Studies using single-axis accelerometers showed lower agreement with reference methods (r = 0.42–0.79), particularly for step length and spatiotemporal variability.

Reliability was analysed in 27 studies (50%), with test–retest reliability examined in 25 studies (93%) and inter- and intra-rater reliability assessed in 2 studies (7%). ICC values for test–retest reliability ranged from 0.53 to 0.95, with higher reliability observed for temporal gait parameters (gait speed, cadence: ICC = 0.80–0.95) than for spatial parameters (step length, stride variability: ICC = 0.53–0.88), and higher values observed in clinical settings (ICC = 0.75–0.95) than in home environments (ICC = 0.53–0.90). Standard error of measurement (SEM) was provided in only two studies (4%) and ranged from 2.1% to 7.8% of the walking distance. Among studies using different smartphone placements, test–retest reliability was highest when devices were positioned at the hip or in the pocket (ICC = 0.85–0.95). Lower reliability was observed in studies placing the smartphone at the chest (ICC = 0.62–0.83) or thigh (ICC = 0.53–0.80), suggesting that placement influences measurement consistency. Longitudinal reliability assessments were conducted in three studies, with monitoring periods ranging from 2 weeks to 6 months. ICC values remained stable over time (ICC = 0.75–0.93) for gait speed and cadence, but showed higher variability for step length and gait variability measures (ICC = 0.58–0.84).

Feasibility was examined in 51 studies (94%), with 40 studies (78%) reporting high participant adherence across different study designs. Clinical studies demonstrated higher completion rates (91%) compared to home-based settings (completion rate 80–85%), where usability challenges were more frequently reported. In 33 studies (61%), minimal training was required before participants could use smartphone-based gait applications. Challenges related to sensor placement inconsistencies were reported in 12 studies (22%), with notable variations observed between hip-, chest-, and pocket-mounted devices. Surface conditions affected measurement quality in nine studies (17%), with gait speed and stride length showing greater variability when measured on soft surfaces, such as carpets, compared to firm flooring. Environmental factors, such as ambient noise interfering with audio-based gait applications, were mentioned in six studies (11%), predominantly in home environments. Among studies evaluating feasibility in older adults, 14 studies highlighted usability concerns related to screen size, touch sensitivity, and software complexity, where 27% of home-based participants required assistance compared to 9% in clinical studies. Participants with motor impairments reported difficulties navigating smartphone applications, particularly when adjusting sensor placement or initiating recordings. Long-term feasibility was examined in 8 studies, where smartphone-based gait monitoring was conducted over periods ranging from one week to six months. Adherence rates remained above 85% in all studies, with the highest retention observed in studies using passive data collection methods (e.g., apps in the background like the IPhone Fitness App or Samsung Health), though home-based engagement declined slightly over time (75–88%) compared to that in clinical settings (>90%).

An overview of all psychometric properties recorded in the individual studies and their methodological overlaps and relationships to each other can be seen in Figure 2.

### 3.4. Methodological Quality of the Included Studies

The methodological quality of each of the included studies is summarised in Table 3. Among them, 87% of the studies (47 studies) met requirements such as clear definitions of the target population and adequate sample sizes. However, notable weaknesses were observed in the blinding of assessors and the use of standardised measurement protocols. Only a small proportion of the studies (approximately 20%) reported fully blinded outcome assessments. Statistical analyses, however, were generally considered appropriate and robust, with a variety of approaches used, including linear models (e.g., regressions or ANOVAs), ICC calculations, and ROC analysis. Consequently, the majority (96%) of the included studies (52 studies) were rated as having “good overall methodological quality” with only 2 studies rated as having “fair overall quality”.

## 4. Discussion

### 4.1. Key Findings

The objective of this systematic review was to critically appraise the psychometric properties of smartphone-based gait analyses across various chronic diseases. Overall, the findings indicate that smartphone applications represent a valid, cost-effective, and accessible alternative to traditional gait analysis methods, such as motion capture systems or force plates, for assessing gait parameters across different chronic diseases, both in clinical and home environments. Additionally, the reliability of smartphone-based gait analysis can generally be rated as good, emphasising the potential of smartphone applications to deliver consistent results. However, it is important to note that reliability has been studied less extensively than validity. Sensitivity and specificity were investigated in 37% of the included studies, with the majority demonstrating high accuracy in distinguishing pathological from healthy gait patterns. Feasibility was confirmed in 94% of the studies, underscoring the widespread acceptance and practicality of these technologies across diverse settings. While these results highlight the potential of smartphone-based gait analysis, several important factors influence its real-world applicability. Variability in measurement accuracy between clinical and home environments, inconsistencies in sensor placement, and the absence of standardised analytical methods all present challenges. Additionally, while validity was high in controlled settings, real-world conditions introduced additional variability that affected measurement consistency. These differences could stem from environmental influences, variations in participant engagement, or limitations in the current algorithms used for gait assessment.

### 4.2. Validity of Smartphone-Based Gait Analysis

The use of smartphone applications has gained remarkable attention over the last couple of years. This is because these devices offer a unique combination of portability, cost efficiency, and accessibility, making them suitable for both clinical assessments and home-based monitoring. Their widespread adoption has been driven by advancements in sensor technology, allowing precise and comprehensive data collection directly from everyday devices. Among the included studies, 70% evaluated validity by comparing results with clinical or biomechanical reference systems. Brooks et al. [43] examined the concurrent validity and demonstrated a high correlation between app-based and clinical walking distance measurements (r = 0.89) in patients with congestive heart failure and pulmonary hypertension aged between 25 and 76 years. Likewise, Polese et al. [58] reported excellent agreement between Google Fit and actual step counts measured with video analyses (ICC = 0.93) in stroke patients aged 62 years. Furthermore, Tang et al. [35] reported strong correlations between a smartphone and the XSens system in Freezing of Gait (FoG) detection (r = 0.86 to 0.97) in Parkinson’s disease patients aged 73 years, reflecting good concurrent validity. Despite these promising results, validity appears to be highly dependent on the specific condition being assessed. Neurodegenerative disorders, such as Parkinson’s disease, showed stronger validity results, likely because gait impairments in these conditions follow clear and well-documented patterns [49]. In contrast, for conditions with more subtle gait abnormalities, such as musculoskeletal or cardiovascular diseases, smartphone-based assessments might be less accurate. This could be because current algorithms may not be sensitive enough to reliably detect these small changes. Additionally, differences in validity between clinical and home environments suggest that external factors impact measurement accuracy [91]. In clinical settings, controlled conditions reduce external variability, leading to stronger validity outcomes (r = 0.75–0.97). However, in home settings, factors such as variations in flooring (e.g., carpet vs. hardwood), lighting conditions, and background noise can create distractions, resulting in inconsistent measurements (r = 0.42–0.91). Another possible explanation is that participants in home environments may use smartphones inconsistently, such as placing them in different pockets or altering their walking speed unconsciously. Future research should explore methods to standardise data collection across different environments, such as real-time feedback to users regarding correct sensor placement.

### 4.3. Reliability of Smartphone-Based Gait Analysis

Reliability was investigated in only 50% of the included studies, with consistently positive results. This indicates that, despite its relevance, reliability was relatively neglected. Bourke et al. [70] reported ICC values ranging from 0.53 to 0.96 for spatiotemporal parameters measured with the FLOODLIGHT software on the smartphone. Creagh et al. [73] demonstrated, for patients with multiple sclerosis aged 40 years, excellent reliability of step counts with an ICC of 0.91. Similar results were reported by Serra-Ano et al. [60], who found ICC values ranging from 0.89 to 0.92 for gait analyses using the FallSkip^®^ app in patients with Parkinson’s disease aged 69 years. In the same sense, Tang et al. [35] reported high reliability in FoG detection (ICC = 0.768–0.896) for persons aged 73 ± 9 years. However, reliability appears to be influenced by several methodological factors. Test–retest reliability was higher in clinical settings (ICC = 0.75–0.95) compared to that in home-based environments (ICC = 0.53–0.90). This discrepancy could stem from differences in walking conditions—hospital corridors provide uniform walking paths, whereas home settings involve more variability in surfaces and obstacles. Another explanation could be user-related factors, such as inconsistent smartphone positioning or fluctuations in walking patterns due to distractions at home. Inter-rater reliability was only assessed in 4% of the studies, highlighting an important research gap. Since smartphone-based gait analysis is designed for both clinical and patient-led use, future studies should investigate how different users (e.g., healthcare professionals vs. patients) influence measurement outcomes. Additionally, long-term reliability remains understudied, with most studies using short-term assessments, leaving uncertainty about whether smartphone-based gait measures remain stable over weeks or months. This is particularly relevant for monitoring progressive diseases, such as Parkinson’s disease, in which gradual changes in gait should be detectable over time.

### 4.4. Sensitivity and Specificity in Pathological Gait Detections

The ability of smartphones to differentiate pathological from healthy populations was examined in 37% of the included studies (20 studies). For example, Arora et al. [69] revealed sensitivity and specificity values of over 90% in distinguishing Parkinsons patients from controls. Similar findings were reported by Tang et al. [35]. Likewise, Mehrang et al. [75] and Pepa et al. [57] demonstrated moderate-to-high accuracy of smartphone algorithms in identifying Parkinson’s disease based on gait parameters, with sensitivity reaching up to 84.9% and specificity up to 95.2%. Additionally, Creagh et al. [73] demonstrated that smartphone-based models could classify gait abnormalities in patients with mild-to-moderate multiple sclerosis with high precision (88%). However, sensitivity and specificity varied between studies, which may be due to differences in the severity of gait impairments among participants. This could suggest that smartphone-based gait analysis may be most effective for detecting moderate-to-severe gait dysfunctions, while its utility for early-stage disease detection requires further validation. Another contributing factor to variability in diagnostic accuracy is the diversity of gait parameters used for classification. Some studies relied on step count and cadence, while others incorporated more complex measures like gait variability and asymmetry. Future research should investigate which combination of gait features yields the highest diagnostic accuracy across different patient populations.

### 4.5. Feasibility and Usability in Clinical and Home Settings

The feasibility of smartphone-based gait analyses was addressed in almost all studies (51 of 54 (94%)), with the majority highlighting their ease of use at home and in clinical settings. Of note, device placement varied considerably across the included studies, from pockets to the hip or chest. While Kim et al. [23] found that the position of the smartphone did not significantly affect FoG detection, Creagh et al. [73] and Brinkløv et al. [89] stressed that inconsistent placements could lead to significant measurement errors. Although feasibility results were generally positive, adherence appears to decline in long-term use. One possible reason is that participants may find repeated gait assessments burdensome or uninteresting over time. Another factor could be technological limitations—while smartphone-based assessments are convenient, individuals with motor impairments may struggle with precise sensor placement or navigating app interfaces. Future research should explore more intuitive designs, such as automatic sensor calibration or voice-guided instructions, to improve usability. Moreover, engagement strategies such as gamification or real-time feedback might enhance adherence, particularly in home-based settings, where user motivation is lower compared to that in clinical environments. Additionally, future studies should investigate whether demographic factors, such as age or digital literacy, influence long-term adherence rates.

## 5. Limitations and Future Research Perspectives

Certain limitations of smartphone-based gait analysis are evident and warrant further investigation. While validity was the primary psychometric feature investigated—reported in 70% of the included studies—reliability received comparatively little attention, with only 27% of studies addressing it. Among these, inter-rater reliability was particularly underrepresented, investigated in just 4% of the studies. This imbalance suggests a need for future research to adopt a more comprehensive approach to psychometric evaluation, ensuring that reliability aspects are adequately explored. In addition, a key challenge lies in the lack of standardisation across studies, particularly regarding device placement, algorithm design, and measurement protocols. Inconsistent methodologies can significantly influence the validity and reliability of findings, limiting their generalisability. Additionally, user experience and satisfaction with smartphone technologies remain underexplored, especially in populations with significant mobility challenges, such as those with advanced motor impairments (e.g., Parkinson’s disease or multiple sclerosis). Future research should address these gaps by prioritising the development of standardised methodologies and refining algorithms tailored to specific diseases and populations.

While the findings highlight the promise of smartphone-based gait analysis, several real-world challenges must be discussed. User compliance remains a significant issue, as individuals with motor impairments, such as Parkinson’s disease or multiple sclerosis, may struggle with smartphone positioning or repeated measurements. Ensuring high user compliance, particularly among older adults and individuals with motor impairments, is essential for the long-term usability of these methods. In this regard, future studies should explore intuitive device handling, personalised feedback mechanisms, and adaptive user interfaces to enhance engagement and reduce technological barriers. Additionally, data security and privacy concerns are crucial barriers to widespread adoption. Many smartphone-based applications rely on cloud storage or external servers, raising ethical concerns about patient data protection. Regulatory frameworks for medical-grade smartphone applications vary across countries, making it difficult to establish uniform standards for data handling and privacy. Another major limitation is the long-term feasibility of smartphone-based gait analysis. While many studies confirm short-term usability [22,23,89], few have examined whether these methods remain reliable over extended periods, or whether users consistently engage with the technology outside of controlled research environments [25,53,59]. Furthermore, environmental factors, such as lighting conditions, surface type, and external interferences (e.g., background noise and sensor drift), can introduce variability in measurements. Addressing these real-world challenges will be critical for ensuring the practical implementation of smartphone-based gait analysis in clinical and home settings. Upcoming research should focus on refining algorithm robustness for different environments, ensuring measurement consistency across diverse conditions, and investigating whether smartphone-based assessments can maintain diagnostic accuracy over time. Additionally, future research should explore ways to enhance usability, compliance, and long-term engagement among patients and clinicians. Moreover, feasibility studies should focus on the needs of diverse user groups to ensure accessibility and practicality in both clinical and home environments. Indeed, integrating smartphone-based gait assessments into routine clinical workflows will be a crucial step toward their widespread adoption. This requires close collaboration among technology developers, healthcare providers, and policymakers to establish standardised guidelines, ensure seamless integration with existing diagnostic tools, and validate their clinical utility in large-scale patient populations. Long-term investigations into the effectiveness of smartphone-based technologies are also needed to evaluate their potential for the personalised treatment and proactive management of chronic conditions. These advancements could promote the widespread adoption of smartphone-based gait analysis tools and enhance their impacts on healthcare delivery.

## 6. Conclusions

This review underscores the growing role of smartphone applications in assessing gait parameters across various chronic conditions. The main findings indicate that smartphone-based gait analyses display good-to-very good validity when compared to gold-standard methods, particularly in controlled environments (i.e., clinical settings). However, measurement accuracy tends to vary in home-based settings due to external factors such as surface variability, sensor placement inconsistencies, and user adherence. Despite these challenges, smartphones offer a practical and scalable alternative for gait monitoring, with significant potential for clinical and remote applications. Overall, the summarised evidence supports their capability to deliver valid, reliable, and sensitive measurements of gait impairments, as well as their utility in distinguishing pathological gait patterns from those of healthy individuals. Reliability was generally rated as good, but inconsistencies in test–retest and inter-rater reliability suggest the need for standardized measurement protocols and long-term validation studies. While sensitivity and specificity were high in certain conditions, such as Parkinson’s disease, variability in diagnostic accuracy across studies highlights the importance of optimising gait parameter selection for different disease populations. Overall, these findings emphasise the potential of smartphone-based gait analysis to effectively contribute to clinical practice and remote monitoring.

## Figures and Tables

**Figure 1 jfmk-10-00133-f001:**
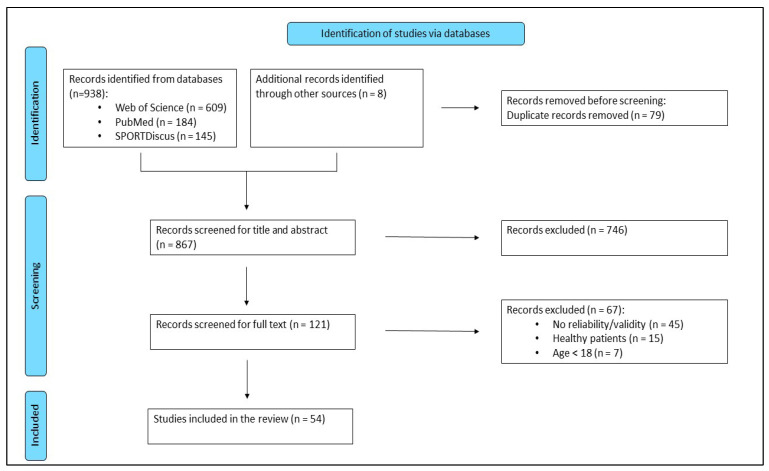
Flowchart of the inclusion and exclusion processes for all studies in the systematic review.

**Figure 2 jfmk-10-00133-f002:**
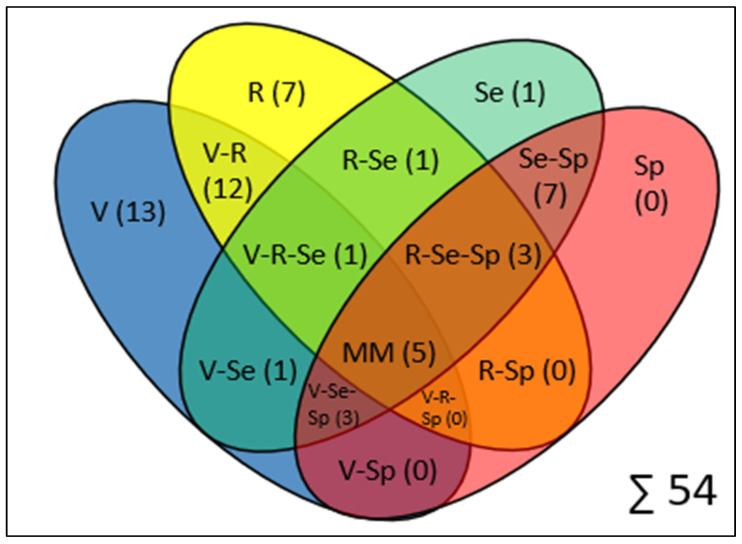
Venn4-Diagram for the overlap of evaluation methods for smartphone-based gait analysis studies, illustrating the numbers of studies assessing validity (V), reliability (R), sensitivity (Se), and specificity (Sp), as well as their methodological overlaps. The numbers of studies are given in brackets as absolute numbers. MM = Multiple Metrics for studies that measured all four psychometric properties.

**Table 1 jfmk-10-00133-t001:** Synthesis matrix of the characteristics of the included studies.

Topic	Main Findings
Total Studies	54 studies included
Study design/Percentage of total studies included	- Cross-sectional studies were the most common/28% - Observational studies were frequently used to analyse real-world feasibility/20% - Longitudinal studies focused on adherence and usability over time/6% - Validation studies directly compared smartphone-based assessments to gold-standard systems/7%
Diseases investigated/Percentage of total studies included	- Parkinson’s disease was the most frequently studied condition/46% - Multiple sclerosis was also widely examined/19% - Stroke (6%), rheumatoid arthritis (4%), and pulmonary diseases (4%) were less commonly analysed - Musculoskeletal conditions (chronic low back pain, orthopaedic disorders, degenerative disc diseases) were assessed in a small subset
Test locations/Percentage of total studies included	- Clinical settings were the primary test locations/52% - Home-based studies explored usability outside of controlled environments/37% - Hybrid studies combined clinical and home settings to examine environmental effects/7%
Smartphone models and Apps/Percentage of total studies included	- Often, iPhones were used for studies, spanning models from iPhone 4s to iPhone 13/26% - Mostly Android devices (Samsung, LG, Xiaomi, Google Nexus) were used/30% - Proprietary apps (mPower, FLOODLIGHT, OneStep, CuPiD, Listenmee, InterWalk) were employed in multiple studies
Sensor placement/Percentage of total studies included	- Hip and pocket placements were the most common, yielding stable recordings/62% - Chest-mounted devices (11%) and thigh/sacrum placements (5%) were investigated as alternatives - Sternum, arm, and lower back placements were used in select studies
Gait parameters analysed/Percentage of total studies included	- Gait speed was the most frequently measured metric/87% - Step length and stride length were analysed in over half of studies/59% - Cadence was commonly assessed, especially in Parkinson’s disease studies/56% - Gait variability and freezing index were examined in neurological populations
Validation methods used/Percentage of total studies included	- Smartphones were validated for gait parameters against motion capture, force plates, or inertial measurement units/65% - Clinical tests, such as the Timed Up and Go, 6-Minute Walk Test, Functional Gait Assessment, and Expanded Disability Status Scale, were used/28% - Some studies reported diagnostic sensitivity and specificity/41%

**Table 2 jfmk-10-00133-t002:** Synthesis matrix of psychometric characteristics of smartphone-based gait analysis across different pathological diseases.

Topic	Main Findings
Total Studies	54 studies included
Reliability/Percentage of total studies included	- Half of the studies assessed reliability/50% - Test–retest reliability was most frequently reported - Inter- and intra-rater reliability were examined in a minority of studies - Intraclass correlation coefficient values ranged from 0.53 to 0.95, depending on study design and gait metric - Strong reliability for gait speed and cadence, but lower for step length and variability
Validity/Percentage of total studies included	- Most studies assessed validity/70% - Concurrent validity was reported/65% - Criterion validity was assessed in half of the studies/52% - Correlations ranged from r = 0.42 to 0.97 - Gait speed and cadence showed stronger correlations (r = 0.78–0.97) - Step length and gait variability showed lower agreement (r = 0.42–0.90)
Sensitivity and Specificity	- Sensitivity ranged from 43% to 98%, and specificity from 59% to 97% - Parkinson’s disease detection (sensitivity 88–98%, specificity 83–97%) showed the highest accuracy - Multiple sclerosis and stroke showed moderate classification accuracy - Musculoskeletal disorders and chronic pain conditions had the lowest diagnostic accuracy due to less distinct gait alterations
Feasibility and Usability/Percentage of total studies included	- Most studies confirmed feasibility/94% - Challenges in long-term adherence were reported, particularly among older adults and those with neurological conditions - Variability in smartphone placement and measurement protocols affected usability outcomes

**Table 3 jfmk-10-00133-t003:** Methodological quality assessment of the included studies.

Author	Question/ Objective	Population	Participation Rate	Selection/ Recruitment	Exposure and Outcome	Timeframe Between Exposure and Outcome	Sample Size	Levels of Exposure	Exposure Measure	Repeated Exposure Measurement	Outcome Measure	Blinding of Outcome Assessors	Follow-Up Rate	Statistical Analyses	Overall Quality
Abujrida et al. [67]	Yes	Yes	Yes	Yes	No	No	No	No	Yes	Yes	Yes	No	No	Yes	Good
Adams et al. [40]	Yes	Yes	Yes	Yes	No	No	Yes	No	Yes	Yes	Yes	No	No	Yes	Good
Alexander et al. [41]	Yes	Yes	Yes	Yes	No	No	No	No	No	No	Yes	No	No	Yes	Fair
Arora et al. [69]	No	Yes	No	Yes	No	No	Yes	No	Yes	No	Yes	No	No	No	Fair
Arora et al. [68]	No	Yes	Yes	Yes	No	No	No	Yes	Yes	Yes	Yes	No	No	No	Fair
Balto et al. [42]	Yes	Yes	Yes	Yes	No	No	No	No	Yes	Yes	Yes	No	No	Yes	Good
Banky et al. [88]	Yes	Yes	Yes	Yes	No	No	Yes	Yes	Yes	Yes	Yes	No	No	Yes	Good
Bourke et al. [70]	Yes	Yes	No	Yes	No	No	No	No	Yes	No	Yes	No	No	Yes	Good
Brinkløv et al. [89]	Yes	Yes	Yes	Yes	No	No	Yes	Yes	Yes	Yes	Yes	No	No	Yes	Good
Brooks et al. [43]	Yes	Yes	Yes	Yes	No	No	Yes	Yes	Yes	Yes	Yes	No	No	Yes	Good
Capecci et al. [44]	Yes	Yes	No	Yes	No	No	Yes	Yes	Yes	Yes	Yes	No	No	Yes	Good
Chan et al. [87]	Yes	Yes	No	Yes	No	No	Yes	No	Yes	Yes	Yes	No	No	Yes	Good
Chen et al. [71]	Yes	Yes	Yes	Yes	No	No	Yes	Yes	Yes	Yes	Yes	No	No	Yes	Good
Cheng et al. [83]	Yes	Yes	No	Yes	No	No	No	No	Yes	No	Yes	No	No	Yes	Good
Chien et al. [46]	Yes	Yes	Yes	Yes	No	No	Yes	Yes	Yes	Yes	Yes	No	No	Yes	Good
Clavijo-Buendía et al. [47]	Yes	Yes	Yes	Yes	No	No	Yes	No	Yes	Yes	Yes	No	No	Yes	Good
Costa et al. [48]	Yes	Yes	Yes	Yes	No	No	Yes	Yes	Yes	Yes	Yes	Yes	No	Yes	Good
Creagh et al. [73]	Yes	No	No	No	Yes	Yes	No	No	Yes	Yes	Yes	No	No	Yes	Good
Creagh et al. [72]	Yes	Yes	Yes	Yes	Yes	Yes	Yes	Yes	Yes	Yes	Yes	No	Yes	Yes	Good
Ellis et al. [49]	Yes	Yes	No	Yes	No	No	Yes	Yes	Yes	No	Yes	No	No	Yes	Good
Ginis et al. [22]	Yes	Yes	Yes	Yes	No	No	Yes	Yes	Yes	No	Yes	No	Yes	Yes	Good
Goñi et al. [80]	Yes	Yes	Yes	Yes	No	No	Yes	Yes	Yes	No	Yes	No	No	Yes	Good
Hamy et al. [85]	Yes	Yes	Yes	Yes	No	No	Yes	Yes	Yes	Yes	Yes	No	No	Yes	Good
He et al. [74]	Yes	Yes	Yes	Yes	No	No	Yes	Yes	Yes	Yes	Yes	Yes	Yes	Yes	Good
Isho et al. [50]	Yes	Yes	No	No	No	No	Yes	No	Yes	Yes	No	No	No	Yes	Fair
Juen et al. [51]	Yes	Yes	Yes	Yes	No	No	Yes	Yes	Yes	Yes	Yes	No	No	Yes	Good
Kim et al. [23]	Yes	Yes	Yes	Yes	No	No	Yes	Yes	Yes	Yes	Yes	No	No	Yes	Good
Lam et al. [84]	Yes	Yes	Yes	Yes	Yes	Yes	Yes	Yes	Yes	Yes	Yes	No	Yes	Yes	Good
Lipsmeier et al. [25]	Yes	Yes	Yes	Yes	No	No	Yes	Yes	Yes	Yes	Yes	No	No	Yes	Good
Lopez et al. [52]	Yes	Yes	Yes	Yes	No	No	Yes	Yes	Yes	No	Yes	Yes	No	Yes	Good
Mak et al. [53]	Yes	Yes	Yes	Yes	No	No	Yes	No	Yes	Yes	Yes	No	No	Yes	Good
Maldaner et al. [54]	Yes	Yes	Yes	Yes	No	No	Yes	No	Yes	No	No	No	No	Yes	Good
Marom et al. [55]	Yes	Yes	Yes	Yes	No	No	Yes	Yes	Yes	No	Yes	No	No	Yes	Good
Mehrang et al. [75]	Yes	Yes	Yes	Yes	No	No	Yes	No	Yes	No	Yes	No	No	No	Good
Omberg et al. [76]	Yes	Yes	Yes	Yes	No	No	Yes	Yes	Yes	Yes	Yes	No	No	Yes	Good
Pepa et al. [56]	Yes	Yes	Yes	Yes	No	No	Yes	Yes	Yes	Yes	Yes	No	No	Yes	Good
Pepa et al. [57]	Yes	Yes	Yes	Yes	No	No	Yes	Yes	Yes	Yes	Yes	No	No	Yes	Good
Polese et al. [58]	Yes	Yes	Yes	Yes	No	No	Yes	Yes	Yes	Yes	Yes	Yes	No	Yes	Good
Raknim et al. [81]	Yes	Yes	No	Yes	Yes	Yes	Yes	Yes	Yes	Yes	Yes	No	Yes	Yes	Good
Regev et al. [59]	Yes	Yes	Yes	Yes	No	No	Yes	Yes	Yes	No	Yes	No	No	Yes	Good
Rozanski et al. [90]	Yes	Yes	Yes	Yes	No	No	Yes	Yes	Yes	Yes	Yes	No	No	Yes	Good
Salvi et al. [86]	Yes	Yes	Yes	Yes	Yes	Yes	No	Yes	Yes	Yes	Yes	No	No	Yes	Good
Schwab et al. [77]	Yes	Yes	Yes	Yes	No	No	Yes	Yes	Yes	No	Yes	No	No	Yes	Good
Serra-Ano et al. [60]	Yes	Yes	Yes	Yes	No	No	Yes	Yes	Yes	Yes	Yes	No	No	Yes	Good
Shema-Shiratzky et al. [61]	Yes	Yes	Yes	Yes	No	No	Yes	No	Yes	Yes	Yes	No	No	Yes	Good
Su et al. [82]	Yes	Yes	Yes	Yes	No	No	Yes	Yes	Yes	Yes	Yes	No	No	Yes	Good
Sugimoto et al. [62]	Yes	Yes	Yes	Yes	No	No	Yes	Yes	Yes	Yes	Yes	No	No	Yes	Good
Tang et al. [35]	Yes	Yes	Yes	Yes	No	No	Yes	No	Yes	Yes	Yes	No	No	Yes	Good
Tao et al. [63]	Yes	Yes	Yes	Yes	No	No	Yes	Yes	Yes	Yes	Yes	No	No	Yes	Good
Van Oirschot et al. [78]	Yes	Yes	Yes	Yes	No	No	Yes	No	Yes	Yes	Yes	No	No	Yes	Good
Wagner et al. [64]	Yes	Yes	Yes	Yes	No	No	Yes	Yes	Yes	Yes	Yes	No	No	Yes	Good
Yahalom et al. [65]	Yes	Yes	Yes	Yes	No	No	Yes	Yes	Yes	Yes	Yes	No	No	Yes	Good
Yahalom et al. [66]	Yes	Yes	Yes	Yes	No	No	Yes	Yes	Yes	Yes	Yes	No	No	Yes	Good
Zhai et al. [79]	Yes	Yes	Yes	Yes	No	No	No	No	Yes	No	Yes	No	No	Yes	Good

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
