# Peer review of "Psychometric Characteristics of Smartphone-Based Gait Analyses in Chronic Health Conditions: A Systematic Review"

_jfmk, 2025, doi:10.3390/jfmk10020133_

Round 1
Reviewer 1 Report
Comments and Suggestions for Authors
Overall, this manuscript is well written. The authors need to address the following comments for its publcation.
Key words need to be changed to reflect the study and omitting reliability and validity is recommended.
In line 37, check a typo error.
In lines 71-80, the authors need to add more explanations to strengthen the need of the study. For example, add some explanations of why psychometric properties are important to be measured, why the comparison of the young and older adults is important, why the understanding of the different chronic conditions is important, etc…..
Measuring feasibility looks arbitrary. Please add more explanations to justify the feasibility measurement.
The discussion is overly optimistic regarding the clinical applicability of smartphones without adequately addressing real-world challenges, such as user compliance, data security, and long-term feasibility. Add some real-world challenges in the discussion.
Author Response
Dear reviewer,
We would like to express our deep gratitude for your valuable time and the constructive and insightful comments you provided. We have addressed your concerns and suggestions in the following point-by-point statement. Whenever necessary, we have made amendments to the manuscript. We sincerely hope you will find our revised manuscript suitable for publication in the Journal of Functional Morphology and Kinesiology.
Kind regards
The authors
Reviewer 1 (Changes highlighted in green)
Comment 1: Overall, this manuscript is well written. The authors need to address the following comments for its publication.
Response: Thank you for your affirmative comment and constructive feedback. Highly appreciated.
Comment 2: Key words need to be changed to reflect the study and omitting reliability and validity is recommended.
Response: Thank you for your comment. Please note that this systematic review focuses on the psychometric properties of smartphone-based gait analysis across various chronic health conditions. As a rule of thumb, keywords should not be redundant with or directly repeated from the title. Instead, they serve a complementary role in enhancing the paper's discoverability in electronic research databases. With this in mind, we used the term “psychometric” in the title from a broader perspective, while adopting a more specific approach in the keywords by including “reliability” and “validity.” Overall, keywords should supplement rather than duplicate the terms used in the title. However, we have added additional keywords to enhance the discoverability and visibility of the work. The following amendments have been made:
“Keywords: Validity; reliability; feasibility; sensitivity; smartphone; inertial measurement units; gait; patients; chronic diseases; pathological conditions; multiple sclerosis; Parkinson diseases”
Comment 3: In line 37, check a typo error.
Response: Thank you for pointing this out. It is now corrected.
Comment 4: In lines 71-80, the authors need to add more explanations to strengthen the need of the study. For example, add some explanations of why psychometric properties are important to be measured, why the comparison of the young and older adults is important, why the understanding of the different chronic conditions is important, etc…..
Response: Thank you for your suggestion. We have adopted your suggestions. The following changes were integrated in the revised version:
“Furthermore, comparing young and older adults is essential, as age-related changes in gait patterns influence the interpretation of smartphone-derived gait metrics. Younger individuals typically exhibit more stable gait patterns, whereas older adults, particularly those with chronic conditions, often present with gait variability, increased fall risk, and mobility limitations. Understanding these differences is fundamental for ensuring that smartphone-based gait assessments are valid across diverse populations. […]
Without robust psychometric evaluation, the clinical applicability of the smartphone-based gait analysis remains uncertain, potentially leading to misleading or non-reproducible results.”
Comment 5: Measuring feasibility looks arbitrary. Please add more explanations to justify the feasibility measurement.
Response: Thank you for that point. We added more explanations.
“In this context, feasibility is a critical factor in determining whether smartphone-based gait analysis can be realistically integrated into routine clinical practice or long-term home monitoring, as technologies with low user compliance or high technical demands may not be sustainable for widespread use (33–35). […]
Additionally, feasibility was assessed based on multiple criteria, including user compliance, ease of use, required technical expertise, and the practicality of smartphone-based gait analysis in different environments (e.g., clinical settings vs. home or remote monitoring). Smartphone-based gait analysis tools were considered feasible if they demonstrated high participant adherence, minimal setup complexity, and usability without extensive training. Of note, feasibility evaluation included qualitative assessments from the original studies, such as reported participant feedback and investigator observations on technical difficulties and measurement consistency.”
Comment 6: The discussion is overly optimistic regarding the clinical applicability of smartphones without adequately addressing real-world challenges, such as user compliance, data security, and long-term feasibility. Add some real-world challenges in the discussion.
Response: Thank you for your legitimate comment. We have added more details related to the potential real-world challenges per the reviewer’s suggestion. The following was integrated into the revised version of the manuscript:
„ While the findings highlight the promise of smartphone-based gait analysis, several real-world challenges must be discussed. User compliance remains a significant issue, as individuals with motor impairments, such as Parkinson’s disease or multiple sclerosis, may struggle with smartphone positioning or repeated measurements. […]
Additionally, data security and privacy concerns are crucial barriers to widespread adoption. Many smartphone-based applications rely on cloud storage or external servers, raising ethical concerns about patient data protection. Regulatory frameworks for medical-grade smartphone applications vary across countries, making it difficult to establish uniform standards for data handling and privacy. Another major limitation is the long-term feasibility of smartphone-based gait analysis. While many studies confirm short-term usability (91, 23, 22), few have examined whether these methods remain reliable over extended periods or whether users consistently engage with the technology outside controlled research environments (76, 53, 59). Furthermore, environmental factors such as lighting conditions, surface type, and external interferences (e.g., background noise and sensor drift) can introduce variability in measurements. Addressing these real-world challenges will be critical for ensuring the practical implementation of smartphone-based gait analysis in clinical and home settings. […]
Additionally, future research should explore ways to enhance usability, compliance, and long-term engagement among patients and clinicians. “
Reviewer 2 Report
Comments and Suggestions for Authors
This study reviews the psychometric properties of smartphone-based gait analysis across 54 papers to determine its validity, reliability, and sensitivity. The results suggest that smartphone-based gait analysis provides effective measurements of gait impairments and can differentiate between pathological and healthy gait patterns. The authors have made considerable efforts to analyze and synthesize the findings from the 54 papers. However, the way to present its importance and outcomes needs further consideration. Specifically, the paper needs a clearer introduction highlighting the importance of studying these psychometric properties. The findings should be described more systematically, and the data presented in Tables 1 and 2 need better organization to enhance readability. It is recommended that the paper be revised thoroughly before resubmission. Therefore, I suggest rejecting this paper in its current form. The authors are recommended to revise the paper accordingly before resubmission.
Please find detailed comments for each section of the study:
Abstract: The abstract is well-structured; however, it is recommended to use "optical motion capture systems" like Vicon for a more accurate reference for the gold standard instead of Xsens, or replace "gold standard" with a more accurate term, as Xsens has known performance issues.
Introduction:
1. Reorganize the first paragraph to clearly show how chronic conditions can lead to gait disturbances and subsequent adverse health outcomes. The current examples given are confusing in terms of correlation or causation among these three aspects. Provide clearer examples to support the initial statement without causing confusion.
2. The study aims to synthesize the psychometric properties of smartphone-based gait analysis; however, the introduction lacks a comprehensive overview of existing gait analysis methods and fails to clearly define and explain the concept of psychometric properties.
3. What do validity, reliability, and sensitivity mean? Definitions of these terms should be included. Additionally, the introduction should explain why validity, reliability, and sensitivity are crucial in gait analysis and describe typical methods for assessing these properties.
4. The authors attempted to identify a research gap, but it is unclear how this gap was determined due to the insufficient background and prior research described in the introduction.
Method:
1. Review the comprehensiveness of the keyword set for pathological conditions and provide criteria for their evaluation.
2. Specify the number of researchers involved in the screening phase.
Results:
1. In the Study Analysis (lines 165-172), the citations from 29 to 36 are very confusing. Please make it clear or delete it.
2. For each paragraph, summarize the studies at a higher level rather than listing conditions one by one.
3. Restructure Table 1 to make it more readable by grouping papers or features and possibly splitting the information into multiple tables. Apply the same approach to Table 2.
4. Provide a clearer summary of the validation, reliability, and feasibility results directly in the text, rather than only in tables.
5. Clarify the details concerning the differences between clinical and home environments, since it is mentioned in the discussion section.
6. It is not clear to me what is useful information in terms of the psychometric properties of smartphone-based gait analyses based on the results. Is it just a summary of different chronic diseases? It may not be enough.
Discussion:
1. Use headings to organize the section and highlight main topics.
2. It is advisable to incorporate visuals, such as diagrams, to illustrate the relationships between findings concerning validity, reliability, sensitivity, and specificity. With 70% of studies evaluating validity, 27% assessing reliability, 41% examining sensitivity, and 35% appraising specificity, it would be beneficial to identify common evaluation methods and any overlaps among these assessments. A diagram could significantly enhance understanding.
3. I am not sure why machine learning is mentioned in the discussion. It is not covered in the previous sections.
4. In my understanding, the current discussion version is not a high-level summary of the findings. Try to offer a deeper interpretation of the results and future challenges and barriers, rather than merely listing the findings.
Conclusion:
1. The conclusion should include key details from the results and discussions that provide clear takeaways for the reader.
Author Response
Dear reviewer,
We would like to express our deep gratitude for your valuable time and the constructive and insightful comments you provided. We have addressed your concerns and suggestions in the following point-by-point statement. Whenever necessary, we have made amendments to the manuscript. We sincerely hope you will find our revised manuscript suitable for publication in the Journal of Functional Morphology and Kinesiology.
Kind regards
The authors
Reviewer 2 (Changes highlighted in yellow)
Abstract:
Comment 1: The abstract is well-structured; however, it is recommended to use "optical motion capture systems" like Vicon for a more accurate reference for the gold standard instead of Xsens, or replace "gold standard" with a more accurate term, as Xsens has known performance issues.
Response: Thank you for your comment. Revised as recommended (Line 18).
Introduction:
Comment 1: Reorganize the first paragraph to clearly show how chronic conditions can lead to gait disturbances and subsequent adverse health outcomes. The current examples given are confusing in terms of correlation or causation among these three aspects. Provide clearer examples to support the initial statement without causing confusion.
Response: Thank you for your valuable suggestion. We revised the first paragraph according to the reviewer’s suggestion. The revised paragraph now reads:
“Chronic health conditions (e.g., Multiple Sclerosis, Parkinson) often lead to gait disturbances, which can significantly impact mobility, independence, and overall quality of life (1, 2). These gait impairments serve as important clinical markers for assessing disease progression and overall health status (3). For example, it is well-established that neurodegenerative diseases like Parkinson’s disease and Multiple Sclerosis can cause motor impairments, leading to slowed walking speed, reduced step length, and increased gait variability (4, 5). Likewise, musculoskeletal disorders such as osteoarthritis and chronic low back pain can result in altered gait mechanics due to pain and joint stiffness, further contributing to mobility restrictions (6, 7). Furthermore, there is evidence that cardiovascular diseases, including peripheral artery disease, have also been associated with gait abnormalities, as reduced circulation can impair muscle function and endurance (8). These gait impairments result in serious health consequences, as they are associated with adverse outcomes such as increased risk of falls, fractures, loss of independence, and mortality (4–7). In this vein, compelling evidence indicates that slower gait speed is a key predictor of frailty and increased mortality risk, while changes in cadence and step length are considered indicators of early neuromuscular or cardiovascular dysfunction (6, 7). For instance, a systematic review and meta-analysis revealed that individuals with persistent low back pain exhibit a significantly slower walking speed and shorter stride length compared to healthy controls, highlighting the functional impact of chronic pain on gait mechanics (8)”
Comment 2: The study aims to synthesize the psychometric properties of smartphone-based gait analysis; however, the introduction lacks a comprehensive overview of existing gait analysis methods and fails to clearly define and explain the concept of psychometric properties.
Response: Thank you for your comment. We included an overview of the existing gait analysis approaches in line with the reviewer's suggestion. There are some more existing gait analysis methods like ultrasonic measurements or measurements with exoskeleton but we didn’t consider them because they are outdated and no longer state of the art. Additionally, there are questionnaires or observations but these methods aren’t objective. Additionally, more details related to the concept of psychometric properties were included. The revised manuscript now reads:
“Traditionally, gait assessment has relied on sophisticated and expensive technologies, including electromyography (9), motion capture systems (e.g., Vicon) (10), force plates and instrumented walkways (11), which are considered the gold standard in gait analysis, typically available only in specialized clinical or research settings (12). While these methods provide accurate measurements of spatiotemporal gait parameters, they require trained personnel and extensive infrastructure, restricting their accessibility and feasibility for widespread clinical use. […]
In recent years, wearable technologies such as inertial measurement units (IMUs), pressure-sensitive insoles, and smartphones have emerged as promising alternative tools (15–17). With embedded sensors such as accelerometers and gyroscopes, smartphones offer a cost-effective, portable, and accessible solution for capturing motion data (18, 19). These devices enable both real-time and long-term monitoring of gait parameters in natural environments, providing insights that are often more representative of a patient’s functional abilities (20). Studies in populations such as individuals with Parkinson’s disease, Multiple Sclerosis and/or healthy older adults have demonstrated the utility of smartphone-based gait analysis (21–27). To ensure that gait analysis methods provide meaningful and reliable data, it is essential to assess their psychometric properties, which include validity and its derivatives, reliability, and sensitivity (28, 29). hese properties are crucial in determining whether a testing method can be trusted for clinical or research applications.”
Comment 3: What do validity, reliability, and sensitivity mean? Definitions of these terms should be included. Additionally, the introduction should explain why validity, reliability, and sensitivity are crucial in gait analysis and describe typical methods for assessing these properties.
Response: Thank you for your comment. The revised version includes definitions of validity, reliability, and sensitivity. Additionally, the importance of these key test properties in the context of gait analysis is now emphasized. Regarding the methods used to assess these properties, we believe they should be included in the methods section, specifically under “data extraction and analysis” section, rather than in the introduction. The manuscript has been revised accordingly as follows:
“From a broader perspective, validity refers to the extent to which a measurement tool accurately assesses what it is intended to measure (30). Concurrent validity, a key validity aspect , refers to the degree to which the measurement tool correlates with a gold standard measure (31). In gait analysis, this means comparing smartphone-based measurements with established gold standard systems, such as motion capture or instrumented walkways. A method with poor validity may produce misleading gait abnormalities. Reliability relates to the consistency and repeatability of a measurement method. A tool is considered reliable if it produces stable and consistent results under similar conditions (26). Sensitivity describes a method’s ability to correctly detect actual gait impairments, while specificity indicates its capability to distinguish between healthy individuals and those with gait abnormalities. High sensitivity ensures that most individuals with gait impairments are correctly identified, reducing the risk of false negatives, whereas high specificity minimizes false positives by correctly classifying healthy individuals as non-impaired (32).”
Comment 4: The authors attempted to identify a research gap, but it is unclear how this gap was determined due to the insufficient background and prior research described in the introduction.
Response: Thank you for the feedback. We believe that the substantial amendments incorporated into the manuscript have adequately addressed this aspect. Of note, we explicitly highlighted the gap in the literature in the last paragraph of the introduction as follows:
“A major gap in the existing literature is the lack of a systematic evaluation of the psychometric properties of smartphone-based gait analysis across different chronic diseases (e.g., Parkinson’s disease, Multiple Sclerosis, and stroke) and age groups. Without robust psychometric evaluation, the clinical applicability of the smartphone-based gait analysis remains uncertain, potentially leading to misleading or non-reproducible results. Therefore, this review aims to fill this gap by critically evaluating the validity, reliability, sensitivity, and feasibility of smartphone-based gait analysis in young and older adults with different chronic conditions. By synthesizing the existing evidence, this review seeks to assess the clinical potential of smartphone-based IMUs for gait monitoring, identify limitations in current research, and propose directions for future studies.”
Methods:
Comment 1: Review the comprehensiveness of the keyword set for pathological conditions and provide criteria for their evaluation.
Response: Thank you for your comment. Please note that this systematic review focuses on the psychometric properties of smartphone-based gait analysis across various chronic health conditions. As a rule of thumb, keywords should not be redundant with or directly repeated from the title. Instead, they serve a complementary role in enhancing the paper's discoverability in electronic research databases. According to the reviewer’s suggestion, we have revised the key keywords as follows:
“Keywords: Validity; reliability; feasibility; sensitivity; smartphone; inertial measurement units; gait; patients; chronic diseases; pathological conditions; multiple sclerosis; Parkinson diseases”
However, we are not sure about the second part of the comment and would highly appreciate providing more details.
Comment 2: Specify the number of researchers involved in the screening phase.
Response: Thank you for your comment. We provided the information in the text in the part “Search Strategy” (Line 141 - 145).
Results:
Comment 1: In the Study Analysis (lines 165-172), the citations from 29 to 36 are very confusing. Please make it clear or delete it.
Response: Thank you for that point. Deleted as recommended.
Comment 2: For each paragraph, summarize the studies at a higher level rather than listing conditions one by one.
Response: Done as suggested. Thank you. The revised manuscript now reads:
“Overall, the studies primarily investigated smartphone-based gait analysis in patients with neurological (e.g., Parkinson’s disease, Multiple Sclerosis, stroke) and musculoskeletal conditions (e.g., rheumatoid arthritis, chronic low back pain). Most studies were conducted in clinical settings (52%), though a significant portion (37%) explored home-based assessments, underscoring the feasibility of smartphone gait monitoring outside traditional healthcare facilities.
The range of smartphone models used in the studies was diverse. 14 studies (26%) employed iPhone models, including older devices like the iPhone 4s (43) and newer models like the iPhone 13 (63). Eight studies (15%) used specialized apps like the mPower app (40, 77), while 16 studies (30%) relied on standard apps such as Google Fit or proprietary applications, as seen in the work from Polese et al. (58). Sensor placement varied considerably across studies, reflecting ongoing methodological debates regarding optimal positioning. Most of the studies (62%) placed smartphones at the hip or in the pocket, aligning with traditional IMU-based gait assessments. Other studies explored alternative placements, such as the chest (11%) or even less conventional positions like the thigh or sacrum (4%). These variations highlight the lack of standardization in smartphone-based gait assessments.
Gait speed was measured in 47 studies (87%), including works by Tang et al. (35) and Adams et al. (40). Step length and stride length were a focus in 32 studies (59%), such as Capecci et al. (44) and Mehrang et al. (77). Gait variability and fluctuation width were analysed in 16 studies (30%), including Kim et al. (23) and Pepa et al. (57). Five studies (9%), such as Kim et al. (23), investigated specific parameters like the freezing index. Overall, most studies focused on fundamental spatiotemporal gait parameters such as gait speed, step length, and cadence, as these are widely used to assess mobility impairments in clinical practice. A smaller subset of studies analysed more complex gait characteristics, such as gait variability, which has been proposed as an early indicator of neurodegenerative disease progression. Some studies specifically targeted disease-specific gait markers, such as the freezing index in Parkinson’s disease.”
Comment 3: Restructure Table 1 to make it more readable by grouping papers or features and possibly splitting the information into multiple tables. Apply the same approach to Table 2.
Response: Thanks for this comment. Table 1 presents the characteristics of the included studies, while Table 2 provides an overview of the psychometric properties of smartphone-based gait analysis across different pathological conditions. Given the substantial number of studies included (n = 54), both tables are relatively long. To enhance readability, we have added horizontal and vertical borders to improve clarity. Splitting them into four tables may compromise readability and make it harder to grasp the overall study spectrum. Therefore, we prefer to maintain the current format, as proper formatting per JFMK requirements should resolve any readability concerns. We hope for your understanding.
Comment 4: Provide a clearer summary of the validation, reliability, and feasibility results directly in the text, rather than only in tables.
Response: Thank you for your comment. More details on validity, reliability, and feasibility outcomes has been incorporated into the revised version as recommended.
“Concurrent validity was the most frequently examined aspect, with 35 studies (65%) comparing smartphone-derived gait parameters to gold-standard systems such as motion capture, force plates, instrumented walkways, or wearable IMUs. Correlation coefficients (r) for concurrent validity ranged from 0.42 to 0.97, with higher agreement levels observed for gait speed (r = 0.78–0.97) and cadence (r = 0.74–0.95) compared to step length (r = 0.42–0.90) and gait variability (r = 0.56–0.87). ICC values for agreement varied from 0.53 to 0.96, depending on the gait metric and study design. Concurrent validity was generally higher in clinical settings (r = 0.68–0.92) compared to home-based assessments (r = 0.50–0.81). This trend was also reflected in the broader comparison of measurement conditions, where differences between clinical and home environments were observed in 37% (20 studies). Specifically, clinical studies showed stronger correlations (r = 0.75–0.97) due to standardized test conditions, whereas home-based assessments exhibited greater variability (r = 0.42–0.91), likely influenced by environmental factors such as surface conditions and participant adherence.
Sensitivity and specificity values followed a similar trend, with clinical assessments demonstrating higher diagnostic accuracy (sensitivity 85–98%, specificity 79–97%) compared to home settings (sensitivity 62–91%, specificity 59–94%). Construct validity, more specifically discriminative validity, was examined in 6 studies (11%), primarily through subgroup comparisons of healthy and pathological gait patterns. Sensitivity and specificity were reported in 41% and 35% of the studies, respectively. Sensitivity values ranged from 43% to 98%, and specificity from 59% to 97%, with the highest classification accuracy reported for Parkinson’s disease (sensitivity 88–98%, specificity 83–97%). Studies on Multiple Sclerosis and stroke showed moderate classification performance, with sensitivity values ranging from 62% to 89%. The lowest diagnostic accuracy was observed in musculoskeletal disorders and chronic pain conditions, where gait alterations were less distinct. Among studies focusing on different smartphone models, validation outcomes varied depending on sensor specifications, with higher correlations observed in studies using devices with tri-axial accelerometers and gyroscopes. Studies using single-axis accelerometers showed lower agreement with reference methods (r = 0.42–0.79), particularly for step length and spatiotemporal variability.
Reliability was analysed in 27 studies (50%), with test-retest reliability examined in 25 studies (93%) and inter- and intra-rater reliability assessed in 2 studies (7%). ICC values for test-retest reliability ranged from 0.53 to 0.95, with higher reliability observed for temporal gait parameters (gait speed, cadence: ICC = 0.80–0.95) than for spatial parameters (step length, stride variability: ICC = 0.53–0.88) and higher values observed in clinical settings (ICC = 0.75–0.95) than in home environments (ICC = 0.53–0.90). Standard error of measurement (SEM) was provided in only 2 studies (4%) and ranged from 2.1% to 7.8% of the walking distance. Among studies using different smartphone placements, test-retest reliability was highest when devices were positioned at the hip or in the pocket (ICC = 0.85–0.95). Lower reliability was observed in studies placing the smartphone at the chest (ICC = 0.62–0.83) or thigh (ICC = 0.53–0.80), suggesting that placement influences measurement consistency. Longitudinal reliability assessments were conducted in 3 studies, with monitoring periods ranging from 2 weeks to 6 months. ICC values remained stable over time (ICC = 0.75–0.93) for gait speed and cadence but showed higher variability for step length and gait variability measures (ICC = 0.58–0.84).
Feasibility was examined in 51 studies (94%), with 40 studies (78%) reporting high participant adherence across different study designs. Clinical studies demonstrated higher completion rates (91%) compared to home-based settings (completion rate 80–85%), where usability challenges were more frequently reported. In 33 studies (61%), minimal training was required before participants could use smartphone-based gait applications. Challenges related to sensor placement inconsistencies were reported in 12 studies (22%), with notable variations observed between hip-, chest-, and pocket-mounted devices. Surface conditions affected measurement quality in 9 studies (17%), with gait speed and stride length showing greater variability when measured on soft surfaces such as carpets compared to firm flooring. Environmental factors such as ambient noise interfering with audio-based gait applications were mentioned in 6 studies (11%), predominantly in home environments. Among studies evaluating feasibility in older adults, 14 studies highlighted usability concerns related to screen size, touch sensitivity, and software complexity, where 27% of home-based participants required assistance compared to 9% in clinical studies. Participants with motor impairments reported difficulties navigating smartphone applications, particularly when adjusting sensor placement or initiating recordings. Long-term feasibility was examined in 8 studies, where smartphone-based gait monitoring was conducted over periods ranging from one week to six months. Adherence rates remained above 85% in all studies, with the highest retention observed in studies using passive data collection methods (e.g., apps in the background like the IPhone Fitness App or Samsung Health), though home-based engagement declined slightly over time (75–88%) compared to clinical settings (>90%).”
Comment 5: Clarify the details concerning the differences between clinical and home environments, since it is mentioned in the discussion section.
Response: Thank you for your time and feedback. The difference between clinical and home environments is now clarified in the “Validation, Reliability and Feasibility Outcomes section”. The following changes have been made:
“Concurrent validity was the most frequently examined aspect, with 35 studies (65%) comparing smartphone-derived gait parameters to gold-standard systems such as motion capture, force plates, instrumented walkways, or wearable IMUs. Correlation coefficients (r) for concurrent validity ranged from 0.42 to 0.97, with higher agreement levels observed for gait speed (r = 0.78–0.97) and cadence (r = 0.74–0.95) compared to step length (r = 0.42–0.90) and gait variability (r = 0.56–0.87). ICC values for agreement varied from 0.53 to 0.96, depending on the gait metric and study design. Concurrent validity was generally higher in clinical settings (r = 0.68–0.92) compared to home-based assessments (r = 0.50–0.81). This trend was also reflected in the broader comparison of measurement conditions, where differences between clinical and home environments were observed in 37% (20 studies). Specifically, clinical studies showed stronger correlations (r = 0.75–0.97) due to standardized test conditions, whereas home-based assessments exhibited greater variability (r = 0.42–0.91), likely influenced by environmental factors such as surface conditions and participant adherence.
Sensitivity and specificity values followed a similar trend, with clinical assessments demonstrating higher diagnostic accuracy (sensitivity 85–98%, specificity 79–97%) compared to home settings (sensitivity 62–91%, specificity 59–94%). Construct validity, more specifically discriminative validity, was examined in 6 studies (11%), primarily through subgroup comparisons of healthy and pathological gait patterns. Sensitivity and specificity were reported in 41% and 35% of the studies, respectively. Sensitivity values ranged from 43% to 98%, and specificity from 59% to 97%, with the highest classification accuracy reported for Parkinson’s disease (sensitivity 88–98%, specificity 83–97%). Studies on Multiple Sclerosis and stroke showed moderate classification performance, with sensitivity values ranging from 62% to 89%. The lowest diagnostic accuracy was observed in musculoskeletal disorders and chronic pain conditions, where gait alterations were less distinct. Among studies focusing on different smartphone models, validation outcomes varied depending on sensor specifications, with higher correlations observed in studies using devices with tri-axial accelerometers and gyroscopes. Studies using single-axis accelerometers showed lower agreement with reference methods (r = 0.42–0.79), particularly for step length and spatiotemporal variability.
Reliability was analysed in 27 studies (50%), with test-retest reliability examined in 25 studies (93%) and inter- and intra-rater reliability assessed in 2 studies (7%). ICC values for test-retest reliability ranged from 0.53 to 0.95, with higher reliability observed for temporal gait parameters (gait speed, cadence: ICC = 0.80–0.95) than for spatial parameters (step length, stride variability: ICC = 0.53–0.88) and higher values observed in clinical settings (ICC = 0.75–0.95) than in home environments (ICC = 0.53–0.90). Standard error of measurement (SEM) was provided in only 2 studies (4%) and ranged from 2.1% to 7.8% of the walking distance. Among studies using different smartphone placements, test-retest reliability was highest when devices were positioned at the hip or in the pocket (ICC = 0.85–0.95). Lower reliability was observed in studies placing the smartphone at the chest (ICC = 0.62–0.83) or thigh (ICC = 0.53–0.80), suggesting that placement influences measurement consistency. Longitudinal reliability assessments were conducted in 3 studies, with monitoring periods ranging from 2 weeks to 6 months. ICC values remained stable over time (ICC = 0.75–0.93) for gait speed and cadence but showed higher variability for step length and gait variability measures (ICC = 0.58–0.84).
Feasibility was examined in 51 studies (94%), with 40 studies (78%) reporting high participant adherence across different study designs. Clinical studies demonstrated higher completion rates (91%) compared to home-based settings (completion rate 80–85%), where usability challenges were more frequently reported. In 33 studies (61%), minimal training was required before participants could use smartphone-based gait applications. Challenges related to sensor placement inconsistencies were reported in 12 studies (22%), with notable variations observed between hip-, chest-, and pocket-mounted devices. Surface conditions affected measurement quality in 9 studies (17%), with gait speed and stride length showing greater variability when measured on soft surfaces such as carpets compared to firm flooring. Environmental factors such as ambient noise interfering with audio-based gait applications were mentioned in 6 studies (11%), predominantly in home environments. Among studies evaluating feasibility in older adults, 14 studies highlighted usability concerns related to screen size, touch sensitivity, and software complexity, where 27% of home-based participants required assistance compared to 9% in clinical studies. Participants with motor impairments reported difficulties navigating smartphone applications, particularly when adjusting sensor placement or initiating recordings. Long-term feasibility was examined in 8 studies, where smartphone-based gait monitoring was conducted over periods ranging from one week to six months. Adherence rates remained above 85% in all studies, with the highest retention observed in studies using passive data collection methods (e.g., apps in the background like the IPhone Fitness App or Samsung Health), though home-based engagement declined slightly over time (75–88%) compared to clinical settings (>90%).”
Comment 6: It is not clear to me what is useful information in terms of the psychometric properties of smartphone-based gait analyses based on the results. Is it just a summary of different chronic diseases? It may not be enough.
Response: The results section, particularly the subsection on psychometric properties, has been substantially revised, with detailed outcomes for each psychometric aspect provided in the updated version. To ensure clarity, we have structured the results explicitly around the key psychometric properties:
Validation outcomes, including criterion validity, more specifically concurrent validity (correlation with gold-standard methods) and discriminative validity (the ability to differentiate between healthy and pathological gait patterns).
Reliability measures, covering test-retest reliability, inter- and intra-rater reliability, and consistency across different settings and sensor placements.
Summary of the feasibility outcomes in terms of participants’ adherence, user-friendliness and environmental challenges in the clinical and home environment.
These findings are crucial for determining the accuracy, consistency, and practicality of smartphone-based gait analyses. We hope this clarification resolves any concerns, but we would be happy to further refine the explanations if needed.
Discussion:
Comment 1: Use headings to organize the section and highlight main topics.
Response: Thank you for that helpful advice. We added headings. Please, refer to the revised version of the discussion (Lines 366; 387; 419; 444; 463).
Comment 2: It is advisable to incorporate visuals, such as diagrams, to illustrate the relationships between findings concerning validity, reliability, sensitivity, and specificity. With 70% of studies evaluating validity, 27% assessing reliability, 41% examining sensitivity, and 35% appraising specificity, it would be beneficial to identify common evaluation methods and any overlaps among these assessments. A diagram could significantly enhance understanding.
Response: Thank you for your feedback. We have integrated the suggested diagram and added it to the results section for a better illustration (Lines 337 - 343).
Comment 3: I am not sure why machine learning is mentioned in the discussion. It is not covered in the previous sections.
Response: Thank you for your comment. This was deleted
Comment 4: In my understanding, the current discussion version is not a high-level summary of the findings. Try to offer a deeper interpretation of the results and future challenges and barriers, rather than merely listing the findings.
Response: We appreciate your comment. We made substantial revisions to the discussion section, providing a more in-depth interpretation of the results. (Lines 379 – 386; 402 – 418; 429 – 443; 453 – 462; 469 – 480; 501 – 505; 518 – 521; 524 - 529).
Conclusion:
Comment 1: The conclusion should include key details from the results and discussions that provide clear takeaways for the reader.
Response: Thank you for that advice. We revised the conclusion by adding more detailed and clearer takeaways for the readers as follows:
“This review underscores the growing role of smartphone applications in assessing gait parameters across various chronic conditions. The main findings indicate that smartphone-based gait analysis display good-to-very good validity when compared to gold-standard methods, particularly in controlled environments (i.e., clinical settings). However, measurement accuracy tends to vary in home-based settings due to external factors such as surface variability, sensor placement inconsistencies, and user adherence. Despite these challenges, smartphones offer a practical and scalable alternative for gait monitoring, with significant potential for clinical and remote applications. Overall, the summarised evidence supports their capability to deliver valid, reliable, and sensitive measurements of gait impairments, as well as their utility in distinguishing pathological gait patterns from those of healthy individuals. Reliability was generally rated as good, but inconsistencies in test-retest and inter-rater reliability suggest the need for standardized measurement protocols and long-term validation studies. While sensitivity and specificity were high in certain conditions such as Parkinson’s disease, variability in diagnostic accuracy across studies highlights the importance of optimizing gait parameter selection for different disease populations. Overall, these findings emphasize the potential of smartphone-based gait analysis to effectively contribute to clinical practice and remote monitoring.”

Round 2
Reviewer 2 Report
Comments and Suggestions for Authors
The authors have done an great job addressing most of my concerns, thank you.
However, I still believe the extremely long tables may impact readability, and not all aspects are fully covered within them. I suggest simplifying the tables and/or using a synthesis matrix tables instead in along with the manuscript. Additionally, consider moving the three large tables to the Appendix.
In addition, the abstract in the submission system is not updated. It still has Xsens as the golden-standard system.
Furthermore, there are some formatting errors. For example, in the sentence: "This study has been registered on the Open Science Framework (OSF) at https://doi.org/10.17605/OSF.IO/QFU8D.Search Search strategy", the phrase "Search strategy" should be removed. Please proofread the entire manuscript for formatting inconsistencies.
Author Response
Reviewer 2 (Changes highlighted in yellow)
Comment 1: The authors have done an great job addressing most of my concerns, thank you.
Response: Thank you for your affirmative comment. Very much appreciated
Comment 2: However, I still believe the extremely long tables may impact readability, and not all aspects are fully covered within them. I suggest simplifying the tables and/or using a synthesis matrix tables instead in along with the manuscript. Additionally, consider moving the three large tables to the Appendix.
Response: Thank you for your comment. We created two new synthesis-matrix tables in line with your suggestion (see Table 1 and Table 2) and the old ones were transferred to the appendix.
Comment 2: In addition, the abstract in the submission system is not updated. It still has Xsens as the golden-standard system.
Response: Thank you for your comment. Please note that this was changed in the revised submitted version. However, we cannot change the first version of the abstract. This is reserved exclusively for the editor. The revised statement in the abstract now reads:
“Validity was evaluated in 70% of the included studies, with results showing moderate-to-strong associations between smartphone apps and gold-standard systems (e.g., Vicon), particularly for parameters such as gait speed and stride length (e.g., r = 0.42–0.97).”
Comment 3: Furthermore, there are some formatting errors. For example, in the sentence: "This study has been registered on the Open Science Framework (OSF) at https://doi.org/10.17605/OSF.IO/QFU8D. Search Search strategy", the phrase "Search strategy" should be removed. Please proofread the entire manuscript for formatting inconsistencies.
Response: Thank you for your comment. Everything has been double-checked and any inconsistencies in the formatting have been corrected.
Round 3
Reviewer 2 Report
Comments and Suggestions for Authors
Thanks for the authors, Table 1 and 2 are much more clear now. Please modidy Table 3 as well. The paper is fine for publication once Table 3 is simplified as well.
Author Response
Comment: Thanks for the authors, Table 1 and 2 are much more clear now. Please modify Table 3 as well. The paper is fine for publication once Table 3 is simplified as well.
Response: Thank you for your constructive feedback, which has contributed to the quality of this work. We would also like to thank you for recognizing the improvements made to Tables 1 and 2 and appreciate your suggestion to simplify Table 3 as well. Specifically, Table 3 presents the methodological quality assessment of the included studies, which is a crucial element of our systematic evaluation. The table follows a structured approach using a modified version of the "Quality Assessment Tool for Observational Cohort and Cross-Sectional Studies" developed by the National Institutes of Health. This tool consists of 14 criteria, each rated as "Yes" or "No," and studies are categorized into three quality levels (good, fair, or low) based on predefined thresholds. Given the relevance of transparently reporting the methodological quality of the included studies in the main text, we believe that further simplification of Table 3 may compromise the clarity and integrity of this assessment. Nevertheless, to enhance readability without omitting key information, we have reviewed the table format and refined its layout where possible. However, if you have specific aspects in mind that could be adjusted while maintaining the rigor of our quality assessment, we would be happy to consider further refinements. We greatly appreciate your careful review of our manuscript and hope for your understanding.